

# A comparison of food sources of nudibranch mollusks at different depths off the Kuril Islands using fatty acid trophic markers

Anatolii Komisarenko[1], Vladimir Mordukhovich[2,3], Irina Ekimova[4] and Andrey Imbs[1]

[1] Laboratory of Comparative Biochemistry, A.V. Zhirmunsky National Scientific Center of Marine Biology, Far Eastern Branch, Russian Academy of Sciences, Vladivostok, Russian Federation

[2] Laboratory of Dynamics of Marine Ecosystems, A.V. Zhirmunsky National Scientific Center of Marine Biology, Far Eastern Branch, Russian Academy of Sciences, Vladivostok, Russian Federation

[3] Department of Ecology, Far Eastern Federal University, Vladivostok, Russian Federation

[4] Department of Invertebrate Zoology, Lomonosov Moscow State University, Moscow, Russian Federation

## ABSTRACT

Gastropod molluscs such as nudibranchs are important members of deep-sea benthic ecosystems. However, data on the trophic ecology and feeding specialization of these animals are limited to date. The method of fatty acid trophic markers (FATM) was applied to determine the dietary preferences of nudibranchs off the Kuril Islands. Fatty acid (FA) compositions of *Dendronotus* sp., *Tritonia tetraquetra*, and *Colga pacifica* collected from deep waters were analyzed and compared with those of *Aeolidia papillosa* and *Coryphella verrucosa* from the offshore zone. The high level of FATM such as 22:5n-6 and $C_{20}$ monounsaturated FAs indicated that *Dendronotus* sp. preys on sea anemones and/or anthoathecates hydroids similarly to that of shallow-water species *A. papillosa* and *C. verrucosa*. The high percentage of tetracosapolyenoic acids and the ratio 24:6n-3/24:5n-6 indicated that *T. tetraquetra* preys on soft corals such as *Gersemia* and/or *Acanella* at a depth of 250 m, but soft corals of the family Primnoidae may be the main item in the diet of *T. tetraquetra* at a depth of 500 m. The high content of Δ 7,13-22:2 and 22:6n-3 shows that *C. pacifica* can feed on bryozoans. In *C. pacifica*, 22:5n-6 may be synthesized intrinsically by the mollusks, whereas odd-chain and branched saturated FAs originate from associated bacteria.

# INTRODUCTION

Nudibranchs are a group of marine soft-bodied gastropod mollusks (Gastropoda: Nudibranchia Cuvier, 1817). The greatest diversity of nudibranchs is observed in warm shallow waters, although nudibranchs occur worldwide, from Arctic to Antarctic regions, with some species discovered at a depth near 2,500 m (*Ekimova et al., 2015*; *Bertsch, 2020*). Identification of food sources of nudibranchs is important for understanding their ecology and description of trophic interactions in marine benthic ecosystems (*Ekimova et al.,*

Corresponding author
Anatolii Komisarenko,
komisarenko.anatoly@gmail.com

*2019b*). Nudibranchs are mostly carnivorous, but detritus and microalgae may comprise some part of their diet (*Ekimova, Deart & Schepetov, 2019a*; *Ekimova et al., 2019b*). Nudibranch can feed on soft corals, reef-building corals, sponges, bryozoans, tunicates, barnacles, sea anemones, jellyfish, ophiuroids, colonial hydroids, and other nudibranchs (*Barnes & Bullough, 1996*; *McDonald & Nybakken, 1997*; *McDonald & Nybakken, 1999*; *Goodheart et al., 2017*). Many nudibranch species exhibit high dietary specialization (*Hoover et al., 2012*; *Goodheart et al., 2017*; *Ekimova et al., 2019b*; *Imbs & Grigorchuk, 2019*; *Mikhlina et al., 2018*; *Mikhlina, Ekimova & Vortsepneva, 2020*). In contrast to shallow-water species, data on feeding regimes of deep-sea nudibranch species still remain limited (*Chimienti et al., 2020*).

Fatty acids (FAs) have been used as biochemical markers to trace predator–prey relationships in marine ecosystems for more than 40 years (*Budge, Iverson & Koopman, 2006*; *Kelly & Scheibling, 2012*; *Braeckman et al., 2015*; *Calado & Leal, 2015*). The method of FA trophic markers (FATM) was already successfully applied to determine possible origins of food in several nudibranch species from tropical shallow waters (*Zhukova, 2014*) and the deep-sea nudibranchs *Tritonia tetraquetra* (Pallas, 1788), *Dendronotus* sp., and *D. robustus* A.E. Verril, 1870 collected in the Kurile Islands region (*Imbs, 2016*; *Imbs & Chernyshev, 2019*; *Imbs & Grigorchuk, 2019*). FATM showed that *Dendronotus* sp. and *T. tetraquetra* prey on different species of cold-water soft corals, while *D. robustus* may consume hydrocorals and bryozoans (*Imbs & Grigorchuk, 2019*). The difference in food sources between these two species belonging to the same genus (*Dendronotus*) and inhabiting the same waters was detected by using FATM. The detection of large amounts of dietary FAs in *T. tetraquetra* (*Imbs, 2016*; *Imbs & Chernyshev, 2019*) showed that the FATM method could be successfully apply for the study of trophic ecology of cold-water nudibranchs.

Waters around the Kuril Islands, with their significant depth differences, are one of the world's most productive marine ecosystem (*Shuntov, Ivanov & Dulepova, 2019*). Nudibranchs are a common animal group of this area and, therefore, play an important role on trophic dynamics in the ecosystem studied. To expand our knowledge on trophic ecology of deep-sea mollusks, FA composition of total lipids of three nudibranch species (*Colga pacifica* (Bergh, 1894), *Tritonia tetraquetra*, and *Dendronotus* sp.) collected from deep waters (up to 500 m) were analyzed and compared with those of two nudibranch species (*Aeolidia papillosa* (Linnaeus, 1761) and *Coryphella verrucosa* (M. Sars, 1829)) from the offshore zone (about 20 m) of the Kurile Islands. Dietary preferences of these five species were studied using the method of FATM. A possible influence of depth on nudibranch feeding specialization was discussed.

## MATERIALS AND METHODS

### Sample collection

Sampling was conducted aboard the R/V Akademik Oparin near Simushir Island (Kuril Islands, Sea of Okhotsk, 47°08′N, 152°14′E) in July 2019. In total, 18 nudibranchs were sampled. Three specimens of *C. verrucosa* and 2 specimens of *A. papillosa* were collected

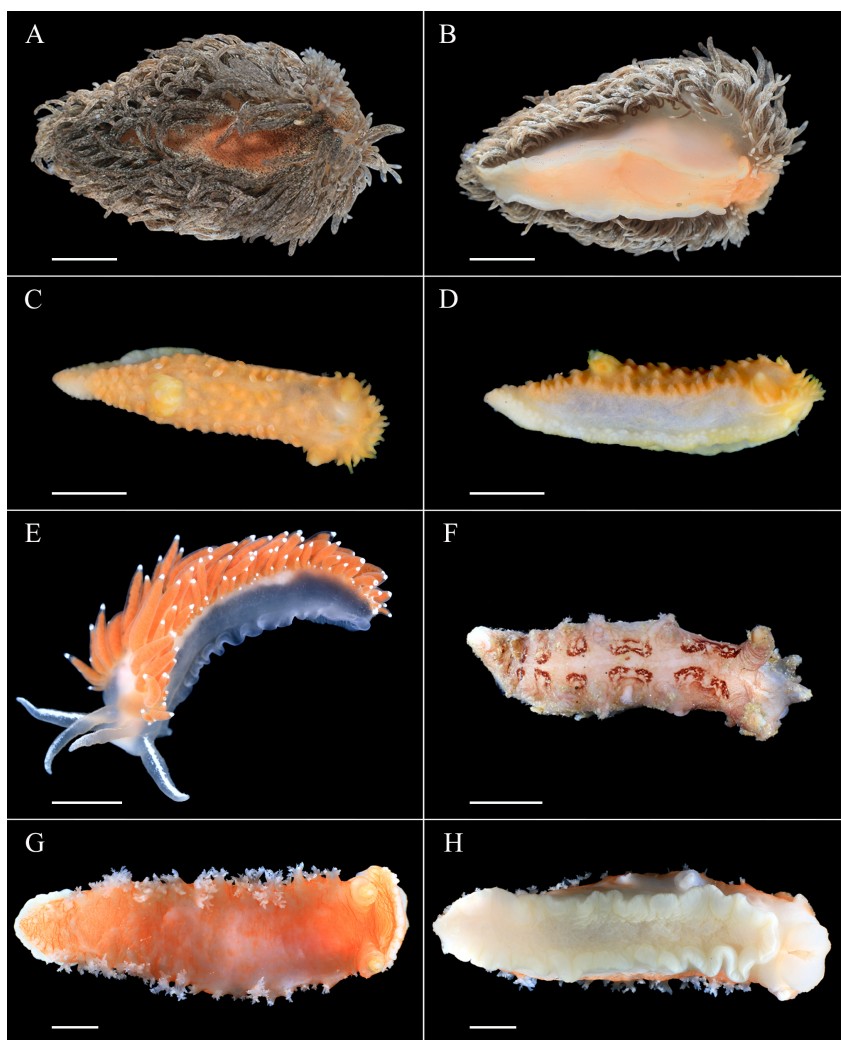

**Figure 1 External views of studied nudibranchs.** (A and B) *Aeolidia papillosa* (dorsal and ventral view, respectively); (C and D) *Colga pacifica* (dorsal and lateral view, respectively); (E), *Coryphella verrucosa*; (F) *Dendronotus* sp.; (G and H) *Tritonia tetraquetra* (dorsal and ventral view, respectively). Scale bar 10 mm. Photos by A. Maiorova.

at a depth of 20 m by SCUBA and referred as a shallow-water group. Three specimens of *C. pacifica*, 3 specimens of *Dendronotus* sp., and 5 specimens of *T. tetraquetra* were collected at the depth of 250–500 m by dredging and referred as a deep-sea group. The nudibranch *Dendronotus* sp. was different from the one already reported (*Imbs & Grigorchuk, 2019*). Each specimen sampled was photographed (Fig. 1); one piece of foot tissue was fixed in 96% EtOH for molecular analysis to confirm the identification of nudibranchs according to *Ekimova et al. (2019b)*; another piece of foot tissue was frozen at −80 °C for lipid analysis. Unfortunately, two frozen samples (one for *A. papillosa* and one for *T. tetraquetra*) were lost in transit.

## Morphological analysis

To study the radular morphology of each species, the buccal mass was extracted and soaked in proteinase K solution for 2 h at 60 °C. When connective and muscle tissues were dissolved, samples were rinsed in distilled water, air-dried, mounted on an aluminium stub, and sputter-coated with gold for visualization under a JEOL JSM 6380 scanning electron microscope. The radular morphology of each species is shown in Fig. 2.

## Lipid preparation and fatty acid analysis

Lipids were extracted from the specimens as described by *Bligh & Dyer (1959)*. FA methyl ethers (FAME) were prepared using the method of *Carreau & Dubacq (1978)* and were purified by preparative thin-layer chromatography in benzene. The 4,4-dimethyloxazoline (DMOX) derivatives of FA were prepared according to the method of *Svetashev (2011)*. A gas chromatography analysis of FAME was conducted with a GC-2010 chromatograph (Shimadzu, Kyoto, Japan) with a flame ionization detector. A Supelcowax 10 (Supelco, Bellefonte, USA) capillary column (30 m $\times$ 0.25 mm ID, film thickness 25 $\mu$m) was held for 2 min at 170 °C, then heated with a 2 °C min$^{-1}$ ramp to 240 °C that was held for 5 min. A sample volume of 1 $\mu$L (about 1 mg mL$^{-1}$) was injected. The injector (250 °C) and detector (260 °C) temperatures were constant. Helium was used as the carrier gas at a linear velocity of 30 cm s$^{-1}$. FAME peaks were analyzed by comparing their retention time with those of the standards (a mixture of PUFA methyl esters No. 3 from menhaden oil, Sigma-Aldrich Co., USA). The concentrations of individual FAs were calculated from the integrated area (% of total FAs). Identification of FAs was confirmed by gas chromatography–mass spectrometry (GC–MS) of their methyl esters and DMOX derivatives on a GCMS-2010 Ultra instrument (Shimadzu, Kyoto, Japan) (electron impact at 70 eV) and a MDN-5s (Supelco, Bellefonte, USA) capillary column (30 m $\times$ 0.25 mm ID). Carrier gas was He at 30 cm s$^{-1}$. The G–MS analysis of FAME was performed at 160 °C with a 2 °C min$^{-1}$ ramp to 240 °C that was held for 20 min. The injector and detector temperatures were 250 °C. GC–MS of DMOX derivatives was performed at 210 °C with a 3 °C min$^{-1}$ ramp to 270 °C that was held for 40 min. The injector and detector temperatures were 270 °C. Spectra were compared with the NIST library and the online FA mass spectra archive website (*Christie, 2021*).

## Statistical analysis

Differences in FA composition (only for species with 3 replicates) were investigated using PERMANOVA (*Anderson, Gorley & Clarke, 2008*; *Clarke & Gorley, 2015*). The PERMANOVA analysis was based on Bray–Curtis similarity matrices, using 9,999 random permutations of raw data. After the PERMANOVA routines, pairwise Monte Carlo tests were performed between all pairs of species. PERMDISP routines was performed to test homogeneity of multivariate dispersions. A nMDS ordination plot was used to visualize the similarity relationship among individuals and groups of individuals. The FAs that characterized and discriminated these groups were identified by SIMPER. The tests mentioned above were carried out using Primer 7+ software (PRIMER-e, New Zealand). Significance of differences in mean contents of FA between the nudibranch

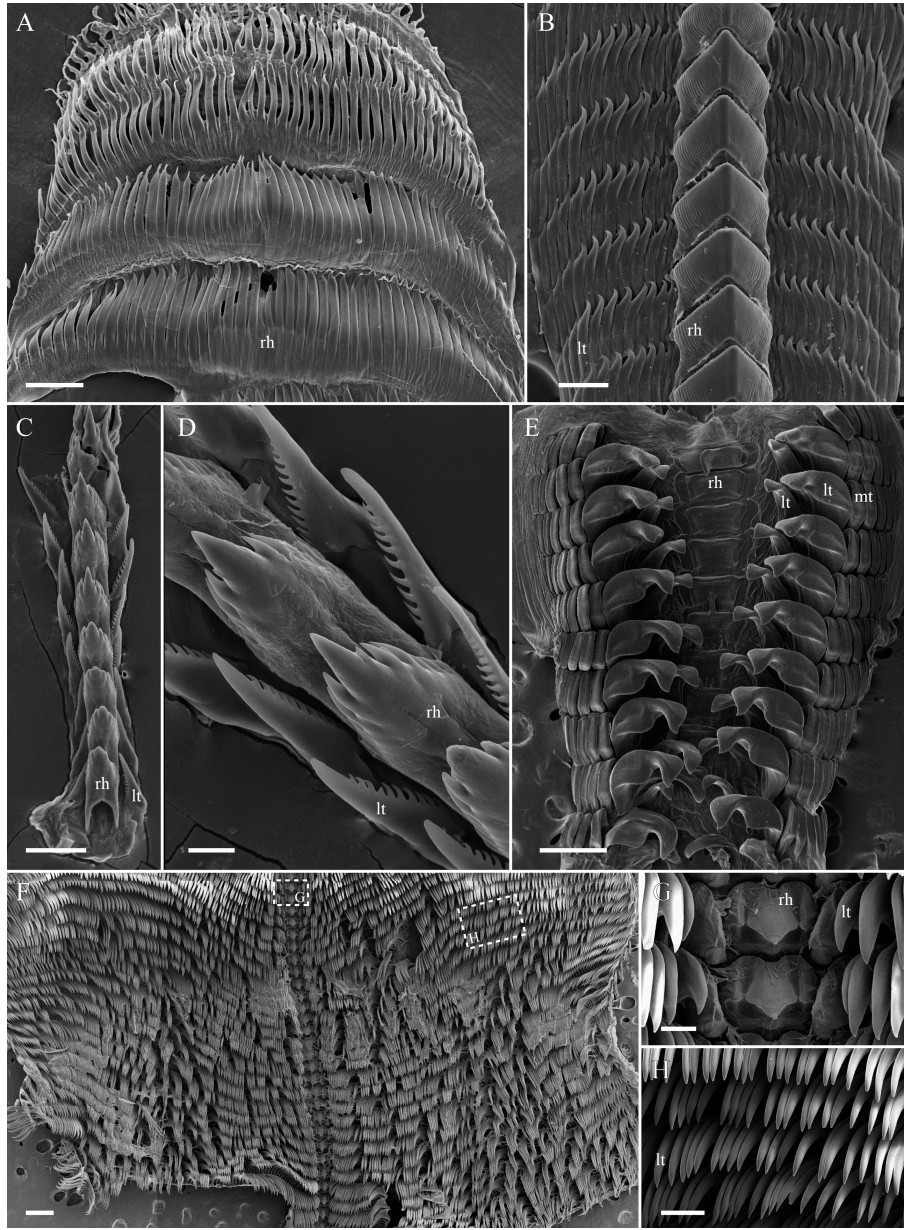

**Figure 2** **Radular morphology of studied nudibranch species.** (A) *Aeolidia papillosa*, posterior portion of uniserial radula. (B) *Dendronotus* sp., middle portion of polycerial radula. (C) *Coryphella verrucosa,* triserial radula. (D) *Coryphella verrucosa,* rachidian and lateral teeth. (E) *Colga pacifica.* (F) *Tritonia tetraquetra,* posterior portion of polyserial radula. (G) *Tritonia tetraquetra,* rachidian teeth and innermost lateral teeth. (H) *Tritonia tetraquetra*, middle lateral teeth. Abbreviations: rh, rachidian tooth; lt, lateral tooth; mt, marginal teeth. Scale bars: A–C, G – 100 μm; D – 30 μm; E, F – 500 μm; H – 200 μm.

species was tested by one-way analysis of variance (ANOVA). Raw data were used after being tested for the homogeneity of variances (Levene's test) and normality of data distribution (Shapiro–Wilk test). Significant differences between levels were examined post hoc with Tukey–Kramer HSD multiple comparisons test. To represent differences

between the nudibranch species, the variables (square roots of FA contents) were included in principal components analyses (PCA). These statistical analyses were performed using STATISTICA 5.1 (StatSoft, Inc., USA). Cluster analysis was performed using Ward's method (Minimum variance method) and the pvclust() function in the pvclust package provides *p*-values for hierarchical clustering based on multiscale bootstrap resampling (*Suzuki & Shimodaira, 2006*) available in the R-Studio software (R-Tools Technology, Canada). A statistical probability of *p* < 0.05 was considered significant. Values are represented as mean ± standard deviation.

## RESULTS

The full FA composition of total lipids in the five nudibranch species from different depths is summarized in Supplement Table S1. The average contents of the major 20 FAs are shown in Table 1. The main saturated FA (SFA) was 16:0, and the major monounsaturated FAs (MUFAs) were 20:1n-9 and 20:1n-7. Lipids of all nudibranchs contained branched and odd-chain SFAs; the highest levels of these acids were detected in some specimens of *A. papillosa* and *C. pacifica* (up to 11 and 17% of total FAs, respectively).

Acids 20:4n-6, 20:5n-3, and 22:6n-3 dominated polyunsaturated FAs (PUFAs) of the nudibranchs studied except for *T. tetraquetra*. The lowest level of 22:6n-3 (HSD test, *p* = 0.0004) and considerable amounts ($F_{4,11}$ = 22.2735, *p* < 0.0001) of very-long-chain tetracosapolyenoic acids (TPA), 24:5n-6 and 24:6n-3, were found in *T. tetraquetra*. The ratio 24:6n-3/24:5n-6 in *Tritonia* specimens from a depth of 450–516 m (7.0 ± 2.2) was higher than that in *Tritonia* specimens from a depth of 210–247 m (1.3 ± 0.1). The level of 20:5n-3 was significantly lower (HSD test, *p* = 0.012) in the deep-sea *C. pacifica* than that in the shallow-water *C. verrucosa*. Unusually high percentages of 22:5n-6 were detected in two specimens of *C. pacifica* (9.6 and 18.6% of total FAs). Individuals of *A. papillosa* contained the highest level of 22:5n-3 (up to 9.9% of total FAs). Several non-methylene-interrupted FAs (NMI FAs) were present in total FAs of all mollusk species. The highest level (HSD test, *p* = 0.0007) of Δ5,11-20:2 in *T. tetraquetra* specimens distinguished them from other nudibranchs. All species (except for *C. verrucosa*) contained noticeable amounts of Δ7,13-22:2.

The PERMANOVA results (Table 2) were corroborated by the nMDS plot (Fig. S1) and revealed significant differences (*p* < 0.001) between species. The pairwise comparison has shown significant (*p* < 0.05) differences for all pairs of species (Table 2), with the exception of the pair *Dendronotus* sp. and *C. verrucosa*. To detail the impact of each FA in similarity and dissimilarity among all nudibranch studied, the FA composition data were analyzed by SIMPER. Table S2 shows the first five FAs that contribute more than 7% in the similarity or dissimilarity. The high level of arachidonic acid (20:4n-6) in *T. tetraquetra* specimens distinguished them from other nudibranchs studied. The level of 20:4n-6 in *Tritonia* specimens from a depth of 450–516 m (10.4–13.6%) was lower than that in *Tritonia* specimens from a depth of 210–247 m (18.5–20.0%) (Table S1).

Analyses of the FA composition data by ANOVA identified certain FAs that were mainly responsible for the difference between species from deep and shallow waters (Table 1).

 

**Table 1 Fatty acid composition (% of total FAs) of nudibranch mollusks.** The species were collected at different depths near Simushir Island (Kuril Islands, Sea of Okhotsk). SFAs, saturated FAs; MUFAs, monounsaturated FAs; PUFAs, polyunsaturated FAs, n-3/n-6, the n-6 PUFAs/ n-3 PUFAs ratio. Values are means ± SD; asterisks indicate significant differences ($p < 0.05$) between the groups of shallow-water species (*C. verrucosa* and *A. papillosa*) and deep-sea species (*C. pacifica*, *T. tetraquetra*, and *Dendronotus* sp).

| Fatty acids | Species names and sampling depths | | | | | Comparison of shallow-water and deep-sea groups by ANOVA | |
|---|---|---|---|---|---|---|---|
| | Shallow-water group | | Deep-sea group | | | | |
| | *Coryphella verrucosa,* 0–20 m, $n = 3$ | *Aeolidia papillosa,* 0–20 m, $n = 2$ | *Colga pacifica,* 285–304 m, $n = 3$ | *Tritonia tetraquetra,* 210–516 m, $n = 5$ | *Dendronotus* sp., 210–516 m, $n = 3$ | $F_{1,14}$ | $p$ |
| 14:0* | 4.2 ± 2.4 | 0.8 ± 0.0 | 1.3 ± 0.6 | 0.4 ± 0.2 | 0.9 ± 0.4 | 7.612 | 0.015 |
| 16:0 | 11.2 ± 2.9 | 7.7 ± 0.9 | 9.3 ± 1.3 | 14.7 ± 2.2 | 12.6 ± 1.7 | 2.456 | 0.139 |
| 16:1n-7* | 2.8 ± 1.0 | 0.7 ± 0.1 | 1.4 ± 0.2 | 0.6 ± 0.1 | 0.8 ± 0.2 | 7.436 | 0.016 |
| 18:0 | 2.7 ± 1.8 | 5.1 ± 0.0 | 3.7 ± 0.4 | 5.5 ± 1.0 | 8.2 ± 1.0 | 3.798 | 0.072 |
| 18:1n-9 | 3.5 ± 1.7 | 1.5 ± 0.1 | 1.3 ± 0.1 | 2.8 ± 0.6 | 2.5 ± 0.6 | 0.495 | 0.493 |
| 18:3n-3 | 1.1 ± 0.2 | 3.5 ± 0.6 | 3.5 ± 1.1 | 0.5 ± 0.1 | 0.8 ± 0.2 | 0.679 | 0.424 |
| 20:1n-11* | 2.8 ± 0.1 | 1.3 ± 0.0 | 1.1 ± 0.1 | 0.6 ± 0.1 | 1.1 ± 0.5 | 6.982 | 0.019 |
| 20:1n-9 | 9.4 ± 2.6 | 1.8 ± 0.1 | 1.9 ± 0.5 | 1.5 ± 0.1 | 4.0 ± 1.1 | 3.053 | 0.102 |
| 20:1n-7* | 5.9 ± 2.3 | 4.3 ± 0.0 | 1.9 ± 0.1 | 2.6 ± 0.2 | 3.9 ± 1.3 | 5.417 | 0.035 |
| Δ5,11-20:2 | 1.3 ± 0.9 | 2.5 ± 0.1 | 0.6 ± 0.2 | 1.6 ± 0.5 | 5.5 ± 0.3 | 0.389 | 0.543 |
| 20:4n-6* | 3.2 ± 0.9 | 3.8 ± 0.2 | 4.3 ± 3.0 | 15.1 ± 4.1 | 8.3 ± 2.5 | 6.290 | 0.025 |
| 20:5n-3* | 25.8 ± 12.4 | 15.5 ± 2.8 | 7.4 ± 3.1 | 13.7 ± 2.8 | 18.4 ± 0.5 | 4.647 | 0.049 |
| Δ7,13-22:2* | 0.6 ± 0.2 | 2.6 ± 0.4 | 10.8 ± 4.7 | 8.8 ± 2.1 | 3.2 ± 1.1 | 11.629 | 0.004 |
| Δ7,15-22:2 | 0.3 ± 0.2 | 1.8 ± 0.4 | 0.7 ± 0.1 | 2.5 ± 0.4 | 1.1 ± 0.4 | 1.249 | 0.282 |
| 22:4n-6* | 3.1 ± 1.8 | 4.1 ± 0.1 | 0.9 ± 0.5 | 0.6 ± 0.1 | 1.6 ± 1.5 | 7.726 | 0.015 |
| 22:5n-6 | 0.1 ± 0.1 | 3.5 ± 0.1 | 9.8 ± 5.9 | 0.6 ± 0.2 | 2.8 ± 1.0 | 0.504 | 0.489 |
| 22:5n-3* | 2.5 ± 1.1 | 8.5 ± 1.9 | 1.4 ± 0.6 | 0.9 ± 0.2 | 2.0 ± 0.5 | 11.292 | 0.005 |
| 22:6n-3 | 8.5 ± 2.1 | 8.3 ± 0.0 | 12.0 ± 3.8 | 0.6 ± 0.1 | 11.0 ± 0.7 | 0.274 | 0.609 |
| 24:5n-6 | 0.0 ± 0.0 | 0.4 ± 0.1 | 0.5 ± 0.4 | 4.8 ± 2.8 | 0.0 ± 0.0 | 2.548 | 0.133 |
| 24:6n-3 | 0.9 ± 0.1 | 0.4 ± 0.0 | 0.7 ± 0.5 | 12.9 ± 2.6 | 0.3 ± 0.1 | 2.933 | 0.109 |
| SFAs | 21.5 ± 2.2 | 22.5 ± 0.9 | 25.3 ± 2.8 | 23.8 ± 2.1 | 25.4 ± 2.1 | 5.951 | 0.029 |
| MUFAs | 27.1 ± 13 | 14.5 ± 0.9 | 12.3 ± 1.4 | 9.7 ± 0.5 | 15.8 ± 1.3 | 7.895 | 0.014 |
| PUFAs | 46.0 ± 16.0 | 44.3 ± 4.4 | 40.5 ± 5.1 | 51.7 ± 3.0 | 44.6 ± 1.5 | 0.104 | 0.752 |
| $n-3/n-6$ | 5.8 ± 1.7 | 2.9 ± 0.3 | 1.1 ± 0.6 | 1.4 ± 0.8 | 2.5 ± 0.3 | 18.998 | 0.001 |

Compared to the group of shallow-water species, deep-sea ones contained significantly higher ($p < 0.05$) levels of 20:4n-6 and Δ7,15-22:2, but significantly lower ($p < 0.05$) levels of 14:0, 16:1n-7, 20:1n-11, 20:1n-7, 20:5n-3, 22:4n-6, and 22:5n-3. No differences ($p > 0.05$) were found for other FAs listed in Table 1.

Results of a cluster analysis of the FA composition data (Table 1) for the five nudibranch species are shown in Fig. 3. All studied specimens were subdivided into three groups: the first and second groups consisted of deep-sea specimens of *T. tetraquetra* and *C. pacifica*,

**Table 2  Results of PERMANOVA pair-wise test of the fatty acid composition of nudibranch mollusks.**

| Groups | t | $p$ (MC) |
|---|---|---|
| CV, CP* | 2.1514 | 0.0296 |
| CV, TS* | 2.6979 | 0.0133 |
| CV, DS | 1.6093 | 0.1118 |
| CP, TS* | 3.4079 | 0.0086 |
| CP, DS* | 2.6239 | 0.0199 |
| TS, DS* | 4.1244 | 0.0024 |

**Notes.**

$p$ (MC) are $p$ values obtained by Monte-Carlo sampling. CV, *C. verrucosa*; CP, *C. pacifica*; DS, *Dendronotus* sp.; TT, *T. tetraquetra*. Asterisks indicates significant differences ($p$ <0.05).

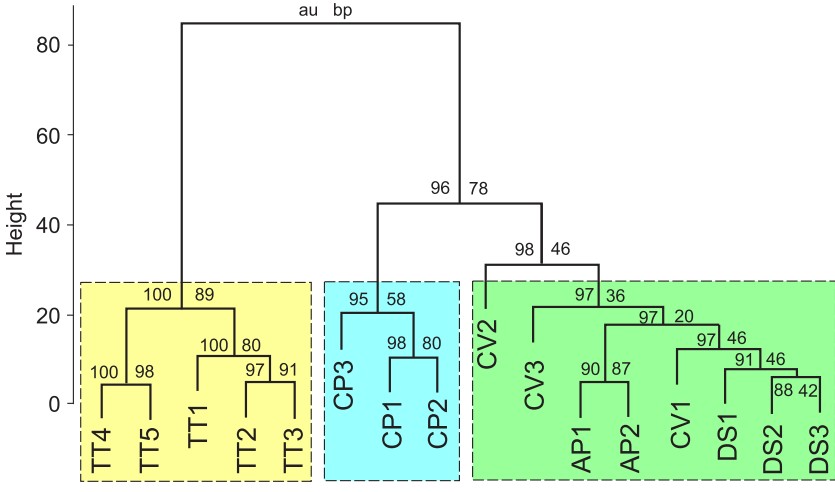

**Figure 3  Results of a cluster analysis of the FA composition data for the five nudibranch species.** The numerals on the branches represent are bootstrap probability (BP) value of a cluster and approximately unbiased (AU) probability values. TT, *Tritonia tetraquetra*; CP, *Colga pacifica*; CV, *Coryphella verrucosa*; DS, *Dendronotus* sp.; AP, *Aeolidia papillosa*.

respectively, and the third group combined specimens of the deep-sea *Dendronotus* sp. with the shallow-water species *A. papillosa* and *C. verrucosa*.

The FAs listed in Table 1 were used as variables for PCA. In this analysis, the first two PCA components explained 50% of the variance of the FA composition data. Figure 4A shows that *T. tetraquetra* is clearly separated from all other nudibranch species along the first PCA component, linking positively with 20:4n-6, 24:5n-6, and 24:6n-3, and negatively with 22:4n-6 and 22:6n-3 (Fig. 4B). The second PCA component separates *C. pacifica* from the group of *Dendronotus* sp., *A. papillosa*, and *C. verrucosa* (Fig. 2A). Figure 2B shows that the level of SFAs (16:0 and 18:0), MUFAs (20:1n-9 and 20:1n-7), and 20:5n-3 *vs.* the level of 22:5n-6 and NMI FAs is significant for this separation. The level of 22:5n-6 was significantly higher ($F_{1,14} = 6.555$, $p = 0.023$) in the group of *Dendronotus* sp., *A. papillosa*, and *C. verrucosa* than that in *T. tetraquetra* and *C. pacifica*. The PCA results (Fig. 4) agree with the results of cluster analysis (Fig. 3) and show a significant difference in FA profiles

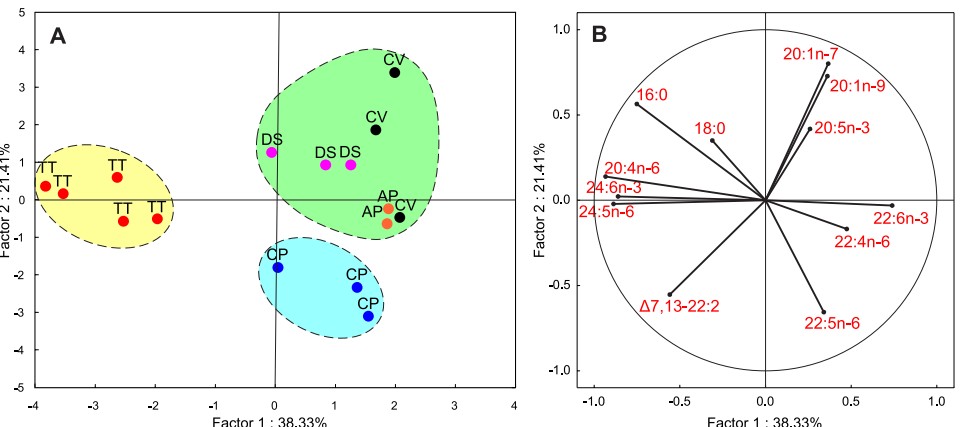

**Figure 4  Results of a principal component analysis (PCA) of the FA composition data for the five nudibranch species.** (A) The plot of the first two principal components; variables were the major fatty acids (see Table 1). Ellipses were drawn manually to outline three groups according to results of the cluster analysis (see Fig. 3). (B) The projectiosn of 12 variables is shown. TT, *Tritonia tetraquetra*; CP, *Colga pacifica*; CV, *Coryphella verrucosa*; DS, *Dendronotus* sp.; AP, *Aeolidia papillosa*.

between deep-sea *T. tetraquetra*, *C. pacifica*, and the two shallow-water species. Both statistical methods confirm that the FA profiles of the deep-sea *Dendronotus* sp. and the shallow-water species are similar.

## DISCUSSION

The representatives of nudibranchs of genera *Aeolidia*, *Coryphella*, and *Dendronotus* prey on various groups of Cnidaria (*Hall & Todd, 1986*). The nudibranch *A. papillosa* is known to prey on sea anemones, grabbing their soft tissues by highly denticulated uniserial radula (Fig. 2A), and consume their nematocyst (stinging capsular organelles) to protect against other predators (*Hall & Todd, 1986*; *Vorobyeva, Malakhov & Ekimova, 2021*). The considerable levels of 22:5n-6 and $C_{20-22}$ MUFAs are characteristic for the FA composition of shallow- and deep-water sea anemones (*Kiyashko et al., 2014*; *Revel et al., 2016*). Obviously, the noticeable amounts of 22:5n-6 and $C_{20}$ MUFAs that we found in *A. papillosa* most likely originate from sea anemone lipids consumed by this nudibranch species.

The nudibranchs *A. papillosa* and *C. verrucosa* occur in the same shallow-water community, but *C. verrucosa* demonstrates non-specified feeding mode (*Mikhlina, Vortsepneva & Tzetlin, 2015*), and its radula (Figs. 2C, 2D) does not take part in biting the prey. This species is known as a non-specialized cnidarian feeder preying on scyphoid jellyfish (*Hernroth & Grondahl, 1985*; *Ostman, 1997*), soft corals (*Sebens, 1983*; *Allmon & Sebens, 1988*), and hydroids of orders Anthoathecata (the genera *Tubularia*, *Clava*, and *Hydractinia*) and Leptothecata (the genus *Obelia*) (*Kuzirian, 1979*). $C_{24}$ PUFAs are proposed as biomarkers for marine food web studies (*Drazen et al., 2008*; *Blanchet-Aurigny et al., 2015*). Very-long-chain $C_{24}$ PUFAs are FATM of jellyfish and soft corals (*Svetashev & Vysotskii, 1998*; *Imbs et al., 2010*; *Imbs, 2016*; *Svetashev, 2019*). Trace amounts of $C_{24}$

PUFAs in *C. verrucosa* indicate that this species from the Kuril Islands probably preys on anthoathecates hydroids, which may be a source of the increasing levels of 22:5n-6 and $C_{20}$ MUFAs in this nudibranch species. *C. verrucosa* is characterized by the least intraspecific similarity in the composition of fatty acids. It may also indicate a wide range of food supplies for this species.

The radula morphology in *Dendronotus* sp. (Fig. 2B) is very similar to that of *Dendronotus lacteus* and *D. rufus* and has a large number of knife-like lateral teeth that nudibranchs may use for biting off soft tissues of polyps (*Ekimova et al., 2019b*). There is some evidence that *D. lacteus* and *D. rufus* feed on hydroids of family Sertulariidae (order Leptothecata), scyphistomaes, and anemones (*Ekimova et al., 2019b*). Considering the close similarity between the FATM profiles of *A. papillosa*, *C. verrucosa*, and *Dendronotus* sp., we assume that the increased 22:5n-6 and $C_{20}$ MUFAs levels recorded in the deep-sea nudibranch *Dendronotus* sp. from the Sea of Okhotsk likely indicate its preying on sea anemones and/or anthoathecates hydroids, similarly to shallow-water species *A. papillosa* and *C. verrucosa*. A dietary resemblance and smoothing of lipid profiles by dietary FAs may be a possible reason of the resemblance in FATM between evolutionary distant species.

Several species of the genus *Tritonia* are known to be obligate predators feeding on soft corals (*Allmon & Sebens, 1988*; *Goddard, 2006*). Recently, an analysis of the FA composition of the nudibranch *T. tetraquetra* preying on soft corals (the Sea of Okhotsk) has shown an intensive transfer of a soft coral FATM (24:5n-6 and 24:6n-3) from prey to predator (*Imbs, 2016*). The ratio 24:6n-3/24:5n-6 was compared between *T. tetraquetra* ($1.1 \pm 0.2$) and several soft coral species. As a result, the soft corals *Gersemia rubiformis* and *Acanella* sp. were suggested as the probable food sources of *T. tetraquetra* (*Imbs, 2016*; *Imbs & Chernyshev, 2019*). No significant differences in the ratio 24:6n-3/24:5n-6 were earlier found between *T. tetraquetra* specimens collected at different depths.

In the present study, the high levels of 24:6n-3 and 24:5n-6, which are observed in *T. tetraquetra* from Simushir Island, confirm preying on soft corals. Based on the ratio 24:6n-3/24:5n-6, we can assume that *T. tetraquetra* at a depth of 250 m mainly feed on the *Gersemia* and/or *Acanella* soft corals. The increase in the ratio 24:6n-3/24:5n-6 accompanying by the decrease in the 20:4n-6 level the in *T. tetraquetra* with increasing depth indicates a change in the taxonomic group of soft corals consumed. Among deep-sea soft corals that occur in the Sea of Okhotsk, the very high ratio 24:6n-3/24:5n-6 = 95 ÷310 and the lowest level of 20:4n-6 ($1.7 \pm 0.3\%$) is characteristic of soft corals within the family Primnoidae (*Imbs, 2016*), which most likely make a considerable contribution in diet of *T. tetraquetra* at a depth of 500 m. Our field observations show that *T. tetraquetra* is often found in communities of various groups of soft corals (Octocorallia) (Fig. S3), which apparently dominate food sources of this nudibranch.

Species of genus *Colga* can feed on members of phylum Bryozoa (*Grischenko & Martynov, 1997*; *Behrens, 2004*). At least 18 species of bryozoans were earlier found in a digestive tract of *C. pacifica* (*Martynov & Baranets, 2002*). A noticeable level of $\Delta$7,13-22:2 and 22:6n-3 has been detected in total FAs of the bryozoan *Dendrobeania flustroides* from the Sea of Okhotsk (*Demidkova, 2010*). The high content of these two FAs in *C. pacifica* confirms that this deep-sea species likely feeds on bryozoans. The low intraspecific similarity in the FA

composition revealed for *C. pacifica* may indicate a lack of food specialization towards any particular species of bryozoan.

Other characteristic FAs of *C. pacifica* such as 22:5n-6 and odd-chain/branched SFAs may originate from own biosynthesis and associated microorganisms, respectively. The unexpectedly high content of 22:5n-6 found in *C. pacifica* may be a result of high activity of $C_2$ elongase and $\Delta 4$ desaturase that convert 20:4n-6 into 22:5n-6. Such activity has been hypothesized in the hydrocoral *Millepora* to explain the extremely high levels of 22:5n-6 and 22:6n-3 (*Imbs, Dang & Nguyen, 2019*; *Imbs et al., 2021*). The relatively low level of 20:5n-3 in *C. pacifica* can be due to either conversion of 20:5n-3 to 22:6n-3 or a deficiency on dietary 20:5n-3 in deep waters (*Kiyashko et al., 2014*). Odd-chain and branched SFAs in marine invertebrates indicate the presence of associated bacteria (*Kharlamenko & Kiyashko, 2018*). Various bacteria have been found in visceral organs of nudibranchs (*Zhukova & Eliseikina, 2012*). An abundant bacterial community may be a cause of the highest level of "bacterial" SFAs in *C. pacifica*.

## CONCLUSIONS

FA profiles of five nudibranch mollusk species belonging to families Polyceridae, Tritoniidae, Dendronotidae, Coryphellidae, and Aeolidiidae were determined. The feeding specialization of deep-sea and shallow-water species were compared on the base of FATM present in their body tissues. Different species originating from different depths, but with similar food sources, showed similar FATM profiles. Species composition of soft corals consumed by *T. tetraquetra* appear to change with increasing depth. Deep-sea nudibranchs of genus *Colga* are most promising objects for future studies, as the proportion between dietary and self-synthesize PUFAs that they feature should be assessed. Future studies employing molecular barcodes to identify nudibranchs gut content can confirm our assumptions on the feeding regimes of the deep-sea species here reported, as FATM provide indirect evidence of trophic interactions and often impair the identification of prey at genus or species level.

### Funding

This work was supported by the Ministry of Science and Higher Education, Russian Federation (grant No. 13.1902.21.0012 for ID, contract No. 075-15-2020-796). The funders had no role in study design, data collection and analysis, decision to publish, or preparation of the manuscript.

### Grant Disclosures

The following grant information was disclosed by the authors:
Ministry of Science and Higher Education, Russian Federation: 13.1902.21.0012, 075-15-2020-796.

## Competing Interests

The authors declare there are no competing interests.

## Author Contributions

- Anatolii Komisarenko performed the experiments, analyzed the data, prepared figures and/or tables, authored or reviewed drafts of the paper, and approved the final draft.
- Vladimir Mordukhovich conceived and designed the experiments, authored or reviewed drafts of the paper, and approved the final draft.
- Irina Ekimova performed the experiments, authored or reviewed drafts of the paper, and approved the final draft.
- Andrey Imbs conceived and designed the experiments, analyzed the data, prepared figures and/or tables, authored or reviewed drafts of the paper, and approved the final draft.

## Data Availability

The raw measurements are available in the Supplementary File.

## Supplemental Information

Supplemental information for this article can be found online at http://dx.doi.org/10.7717/peerj.12336#supplemental-information.

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
