# Peer review of "A comparison of food sources of nudibranch mollusks at different depths off the Kuril Islands using fatty acid trophic markers"

_PeerJ, doi:10.7717/peerj.12336_

## Round 0.1 · original submission · Major Revisions

As you will see the reviewers had positive comments about your manuscript but also raised some concerns. Specifically, they made some suggestions about revising the text and statistical analysis.

·

Basic reporting

The work entitled “A comparison of food sources of nudibranch mollusks at different depths off the Kuril Islands using fatty acid trophic markers” by Anatolii Komisarenko and colleagues is written in a clear way with a suitable level of English (although I suggest some editing in the PDF file I am attaching with my comments) that confirms the relevance of using fatty acids as trophic markers to shed light over the feeding regimes of some deep-sea nudibranchs.
The manuscript is very well organised and easy to follow.
The introduction is concise, but it clearly prepares the reader for the content of the manuscript and provides a good level of background information about the topic, whose scientific content the author team is certainly familiar with (given their solid publication record on this topic).
The literature cited is adequate and up to date (both in the introduction and the discussion).
Figures are clear, but Figure 2 could benefit from being in colour to allow the reader to better discriminate the different species of nudibranchs in the scatter plot, as well as to identify the groups that were formed. If available, the authors could consider supplying colour pictures of the nudibranch species here addressed (everyone loves pictures of these wonderful marine invertebrates).
Table 1 can be improved by listing relevant ratios of some FA useful to discriminate species using FATM, as well as listing the sums of SFA, MUFA, PUFA and HUFA per species (and highlighting significant differences). In Table 1, significant differences should be signalled comparing all species vs. all species and not between shallow vs. deep sea species.
I could not access raw data, only an excel file with Supplementary Table 1 which are not raw data (I please ask the Editorial team to check if taw data is available).

Experimental design

The manuscript submitted is original, clearly within the scope of PeerJ and builds upon previous works of some of the co-authors that have a solid publication record on using FATM to shed light over the trophic regimes of nudibranchs. The manuscript has a well-defined research-question that contributes to advance the state of the art (and the authors clearly explain how they will do so).
While the research is sound, with high technical and ethical standard, there are some important details missing in the methods that require attention by the authors.
In the methods, the authors must detail what was the amount of sample injected for FA analysis? Was any internal standard used? Please clearly refer how fatty acids are expressed (I inferred these were relative abundances of the total pool of FA?) (please see the PDF file I am attaching with my comments).
The authors must explain how they could have performed a one-way ANOVA to compare the FA profile of different nudibranch species, when one of them (Aeolidia papillosa) has n=2? Please clarify this important issue. Performing a one-way PERMANOVA can be an option to compare the FA pool of each species as whole against each other to confirm or refute significant differences. However, n=2 for A. papillosa is a caveat for this and other statistical analysis… Please elaborate on this. Moreover, if raw data were % of FA was it not necessary to transform your data prior to analysis? Please confirm this, as well if the assumptions of ANOVA were always met so there was no need to perform non-parametric statistics (but first pay attention to the fact of n=2 for A. papillosa.

Validity of the findings

While the findings reported are certainly relevant, the authors must clarify the issues referred above concerning their statistical analysis. The authors may also consider suggesting the use of molecular barcodes of gut content of these nudibranchs in future studies to support their assumptions on the feeding regimes of these deep-sea species. The use of FATM provides indirect evidence (and the authors must be cautious when reporting their findings) of trophic interactions and may not be able to discriminate which prey is being ingested to genus or species level.

Additional comments

Given the issue I refer with your statistical analysis I consider that this study requires a major revision before being accepted for publication.
Please see the PDF file I am attaching with my comments.

Reviewer 2 ·

Basic reporting

This is a very interesting paper focused on the little-known topic dealing with the food preference of Nudibranchia. The trophism of five species is studied using the fatty acid trophic markers (FATM) that is poorly used in this group of mollusks. This is a still unexplored field and this work can be an excellent starting point. The English is good and the paper is well structured.

Anyway, I suggest to implement the data on the genera investigated and in general on the biological and ecological context dealing with the species investigated. In fact, I found that the technique used and the results obtained were well explained but the paper does not give the instrument to understand the biological question that is behind this paper. Why is it important to understand the trophism of these species, why and how it is useful the FATM to explore the ecology of these species. I found the paper difficult to follow. Increasing the references cited or reporting data on the state of the art on the knowledge on the diet of related species could be very useful to easily follow the paper and quickly understand the meaning of the study carried out. For example, as little is known to date about the diet of nudibranchs, I find it a bit superficial not to mention all the few works published to date, at least on the genera treated here. I suggested some of these papers directly on the attached pdf file.

Experimental design

It is not clearly reported in the text how many individuals were analyzed and the lack of any kind of pictures of the species investigated make this paper a bit abstract and too methodological. Furthermore, I think that these data should be included also because two of the five analyzed species are not identified at species level but reported as ‘sp.’.

Another point is about the comparison with different species from shallow water as A. papillosa and C. verrucosa. In fact, in the discussion section, you stated that: “Considering the close similarity between the FATM profiles of A. papillosa, C. verrucosa, and Dendronotus sp., we assume that the increased 22:5n-6 and C20 MUFAs levels in the deep-sea nudibranch Dendronotus sp. from the Sea of Okhotsk indicate its preying on sea anemones and/or anthoathecates hydroids, similarly to the shallow-water species, A. papillosa and C. verrucose.” This concept means that the FATM profiles are not so specific for the prey because the three species here discussed (Dendronotus sp., A. papillosa and C. verrucosa) are very distant in terms of evolution and diet and I would aspect to found differences in the FATM profiles. I think that this aspect should be better explained or clarified.

Validity of the findings

I really appreciated to read this paper that is focused on the important, and very hard to study, aspect of the trophism in nudibranchs. The fatty acid trophic markers (FATM) are a consolidated technique but still little used in this group of mollusks today and for this reason this paper is original and interesting. This work can be an excellent starting point to understand and fill the gap of knowledge on this unexplored field of study. Anyway, as I stated above, I think that the authors should make a little extra effort to better explain the biological question behind their work and allow the reader to understand the study done and its possible importance and applications.

Additional comments

Other minor comments are directly reported on the pdf file.

Annotated reviews are not available for download in order to protect the identity of reviewers who chose to remain anonymous.

---

## Round 0.2 · Minor Revisions

The reviewers were satisfied with your changes to the manuscript. One reviewer made some further suggestions for minor revisions that I advise the authors to consider.

·

Basic reporting

The authors have successfully addressed all issues raised on the first version of their study and significantly improved the overall quality of their work. Congratulations!

Experimental design

The number of specimens sampled is now clearly detailed, as well as the methods for FA analysis and statistical comparisons.

Validity of the findings

The findings reported are now much better framed and presented. The discussion of the manuscript is now clearer and easier to follow.

Additional comments

The Figures added to this revised version significantly improve the overall quality of the manuscript. Well done!
I recommend some minor corrections before acceptance of this manuscript (please see the PDF file I am attaching).

Reviewer 2 ·

Basic reporting

No comment

Experimental design

No comment

Validity of the findings

No comment

Additional comments

I thank the authors for the effort in reviewing the manuscript that now resulted to be really improved.
I like the manuscript as it is, and I do not have any other comment.

---

## Round 0.3 · accepted · Accept

Thank you for making the necessary edits to your manuscript.